# Obtaining Precise Molecular Information via DNA Nanotechnology

**DOI:** 10.3390/membranes11090683

**Published:** 2021-09-02

**Authors:** Qian Tang, Da Han

**Affiliations:** 1Institute of Molecular Medicine, Renji Hospital, School of Medicine, Shanghai Jiao Tong University, Shanghai 200127, China; qtang@sjtu.edu.cn; 2School of Materials Science and Engineering, Shanghai Jiao Tong University, Shanghai 200240, China

**Keywords:** DNA nanotechnology, single-molecule techniques, cryo-EM, structural reconstruction, protein–protein interactions, molecular forces

## Abstract

Precise characterization of biomolecular information such as molecular structures or intermolecular interactions provides essential mechanistic insights into the understanding of biochemical processes. As the resolution of imaging-based measurement techniques improves, so does the quantity of molecular information obtained using these methodologies. DNA (deoxyribonucleic acid) molecule have been used to build a variety of structures and dynamic devices on the nanoscale over the past 20 years, which has provided an accessible platform to manipulate molecules and resolve molecular information with unprecedented precision. In this review, we summarize recent progress related to obtaining precise molecular information using DNA nanotechnology. After a brief introduction to the development and features of structural and dynamic DNA nanotechnology, we outline some of the promising applications of DNA nanotechnology in structural biochemistry and in molecular biophysics. In particular, we highlight the use of DNA nanotechnology in determination of protein structures, protein–protein interactions, and molecular force.

## 1. Introduction

Living systems are organic integrations of biomolecules and are controlled by the interactions thereof between each other. Uncovering the structural information of biomolecules and measuring the molecular metrics (i.e., molecular numbers, position, distance, and orientation, as well as the interaction force) can provide important instructions for understanding the mechanisms and functions of biomolecules and guiding the development of related drug discovery. In recent decades, the development of single-molecule techniques such as single-particle cryo-electron microscopy (cryo-EM) [1,2], single-molecule fluorescence resonance energy transfer (FRET) [3], and single-molecule force spectroscopy [4] has expanded the ability to precisely study single biomolecules. Utilizing these single-molecule techniques, researchers are able to reconstruct the structures of biomolecules such as proteins at an atomic-level resolution [2,5], and accurately investigate intermolecular interactions such as protein–protein interactions (PPIs) [6,7] and protein–DNA interactions [8]. These advancements are rapidly transforming our understanding of the structure of biomolecules [9], as well as biological processes [10].

The DNA (deoxyribonucleic acid) molecule is the genetic information carrier for living organisms. At the same time, it is also a building block [11,12,13,14,15,16,17,18,19,20,21] for constructing various nanostructures that are summarized as the significant field of DNA nanotechnology [22,23,24,25]. DNA nanotechnology enables the design and implementation of arbitrarily shaped nanostructures with nanoscale accuracy [26,27,28]. These DNA-based architectures are increasingly involved in various applications in which the accurate addressability and programmability of molecules are needed. In our opinion, the assembly accuracy offers an attractive platform for measuring and recording molecular information such as structures and interactions. Therefore, we summarize the recent advancements in utilizing DNA nanotechnology in this direction, particularly in aiding cryo-EM-based structure determination and analyzing PPIs and PPI-related interaction forces. We anticipate that the interpretation and summary will shed light on more related applications of obtaining precise molecular information via DNA nanotechnology.

## 2. DNA Nanotechnology

DNA (deoxyribonucleic acid) is a polymer made from multiple nucleotides, with each nucleotide containing a deoxyribose, a phosphate group, and one of the four nucleobases (adenine (A), thymine (T), cytosine (C), or guanine (G)). Two separate DNA strands are bound together in a shape of double helix by the hydrogen bond between bases (A with T, C with G), which is known as Watson–Crick base pairing [29]. Specifically, the DNA double helix has a diameter of 2 nm, a helical turn of 10.5 bp (or 3.4 nm), and a persistence length of ~50 nm [30]. In the past few decades, the thermodynamics of DNA hybridization have been well characterized [31,32,33,34,35], and the costs of synthesis and purification of DNA have dramatically decreased. These developments provided the foundation for the field of DNA nanotechnology and have led to great progress in programmable DNA self-assembly and molecular computation.

### 2.1. Structural DNA Nanotechnology

The conceptual foundation of structural DNA nanotechnology can be traced back to the 1980s, when Seeman proposed utilizing DNA structures to aid the crystallization of proteins [36]. Four decades later, the field of structural DNA nanotechnology has delivered enormous capacity to construct static and dynamic nanostructures with unprecedented precision and complexity. In general, there are two main strategies for creating DNA structures: short DNA strand-associated DNA tile assembly and long DNA strands-associated DNA origami assembly. Both strategies are based on the rule of DNA base pairing, also known as Watson–Crick base pairing [29].

DNA tiles are a set of artificial DNA nanostructures composed of several short single-stranded DNA (ssDNA) molecules. Using rational design of sequences with sticky ends, hierarchical assembly of DNA tiles can lead to the formation of larger DNA structures. In 1983, Seeman firstly proposed an immobile four-way junction tile assembled from four different ssDNA strands with sticky ends [37] (Figure 1A).

To construct more stable and rigid DNA tiles, double crossover (DX) tiles were invented [38] (Figure 1B). Taking advantages of the structural rigidity of DX tiles, two-dimensional (2D) DNA crystal structures with diverse patterns were fabricated from three-arm [39] and four-arm DNA tiles [40] (Figure 1C). Subsequently, DNA tiles with more complexity were reported, including triple crossover [43], paranemic crossover [44], DX triangle [45], and ssDNA tiles, the last of which were also called DNA bricks [11,46] (Figure 1D). In the DNA brick strategy, each ssDNA strand acts as a molecular block to interact with others programmably, thus forming designed 2D or 3D objects with much larger sizes.

DNA origami was first developed by Rothemund in 2006, using hundreds of short synthetic oligonucleotides (20–60 nt long, known as staple strands) to fold the M13 phage genomic DNA (a long circular ssDNA strand, typically 7000 nt long, termed a scaffold strand) into various nanoscale shapes [26]. It is worth noting that a few studies proposed similar concepts prior to the DNA origami era. For instance, in 2004, Shih et al. folded a 1.7-kilobase ssDNA strand into a 3D octahedron with the help of five 40 nt ssDNA strands [41] (Figure 1E). Nonetheless, these methods had not demonstrated the versatility for construction of DNA nanostructures with complex and arbitrary geometries. In the first report of DNA origami in 2006, arbitrarily shaped 2D nanostructures with sizes close to 100 nm were designed and implemented using the same design rules.

The milestone of progress in constructing 3D DNA origami was reported by Douglas et al. from the Shih group in 2009 [27]. They constructed 3D objects by bundling DNA helices into a honeycomb lattice (Figure 1F), and developed a design platform named caDNAno [47], which provided a general route and a graphical user interface to create DNA nanostructures. This drastically simplified the design process of DNA origami-based nanostructures and lowers the threshold for making DNA nanostructures. Their subsequent work introduced twisted and curved 3D nanoscale shapes by adding or deleting bases between adjacent DNA helical crossovers [17] (Figure 1G,H). Besides the honeycomb lattice model, square lattice and hybrid lattice design concepts were also proposed [48]. Different from conventional rigid lattice design strategies, the Yan group constructed hollow 3D objects with complex shapes by arranging interhelical angles and lengths of adjacent DNA helices into concentric rings [49]. Later, to overcome the size limitation of less than 0.05 µm^2^ for conventional DNA origami structures, Tikhomirov et al. used multiple DNA origami tiles to assemble DNA origami arrays with sizes up to 0.5 µm^2^, allowing the rendering of images, such as the Mona Lisa, with up to 8704 pixels [42] (Figure 1I).

As for the design strategy of DNA nanostructures, in addition to caDNAno mentioned above, a variety of DNA origami design strategies or software have been developed, such as Tiamat [50] and scadnano [51]. However, the lack of systematic design rules for staple strands and the manual scaffold routing more or less limit the use of these programs, especially for those who are not experts in the field of DNA nanotechnology, but want to utilize DNA nanostructures. Hence, strategies have been developed to realize the automated design of DNA origami [52,53,54,55]. For instance, Benson et al. developed a highly automated method (refer to vHelix, a plugin for Autodesk Maya) of folding arbitrary polygonal digital meshes in DNA [52]. Jun et al. developed PERDIX (Programmed Eulerian Routing for DNA Design using X-overs) for autonomously designing 2D DNA origami by enabling the full autonomy of scaffold routing and staple sequence design with arbitrary network edge lengths and vertex angles [54]. With the help of these tools, researchers can design customized DNA architecture without knowing detailed design principles.

### 2.2. Dynamic DNA Nanotechnology

Dynamic DNA nanotechnology involves the creation of nanoscale DNA devices whose primary function arises from their ability to undergo controlled motion or reconfiguration. Dynamic DNA nanotechnology is based on dynamic DNA regulation methods such as toehold-mediated DNA strand displacement (TMSD), as shown in Figure 2A. TMSD was invented by Yurke et al. in 2000 [56]. By varying the length and sequence composition of toeholds [57], the rate of strand displacement reactions can be quantitatively controlled over a factor of 10^6^ (Figure 2B), which allows engineering control over the kinetics of DNA-based devices. For instance, TMSD has been used for actuating DNA devices, including DNA walkers [58,59], smart drug-releasing nanoboxes [60], diverse types of logic circuits, cascading networks [28,61,62], and molecular computing [28,63,64,65,66].

In addition to TMSD, aptamer-induced binding provides another way for actuation of DNA devices. Aptamers are oligonucleotides that bind specifically to proteins, small molecules, and a range of other targets [67,68,69,70] (Figure 2C). Utilizing the affinity between aptamers and target molecules, dynamic DNA devices have been constructed by designing the actuation part of the devices with incorporation of aptamer sequences [67,71,72]. Upon the reaction with targets, the aptamer can induce an allosteric effect and allow the devices to be actuated. As shown in Figure 2D, Douglas et al. created a DNA nanorobot that can be opened by different combinations of receptors on cell membranes, therefore allowing the relevant applications in diagnostics [71].

Considering that using large molecules such as DNA and proteins to actuate DNA devices is relatively slow kinetically, and often requires minutes to hours to complete the actuation (the rate constant of a TMSD reaction ranges from 10 to 10^6^ M^−1^·s^−1^ [57], while the folding of an acidic pH-triggered i-motif has a rate constant of ~8.35 min^-1^ [73], or takes only ~100 ms time [74]), environmental factors such as pH [75,76,77,78,79,80], light [81,82], temperature [83,84], and ionic concentration [85,86,87] are also used to trigger DNA devices, and have the advantages of being rapidly and remotely controllable. For instance, i-motifs [76,78] and Hoogsteen base pairing [79,80] (Figure 2E) can be used to design devices that are responsive to changes in pH [75,88] (Figure 2F). Light-actuating DNA devices have been created by attaching azobenzene [89] to the DNA backbone, since the azobenzene undergoes isomerization from the *trans* to *cis* conformation upon exposure to UV light and returns to the *trans* when exposed to visible light. Incorporating an ortho-nitrobenzyl moiety [81] (Figure 2G,H) is another approach to constructing a light-actuating DNA device.

## 3. DNA Nanotechnology for Structural Reconstruction of Proteins

Technological breakthroughs in cryo-EM have made it possible to obtain the structures of biological macromolecules and their complexes with near-atomic resolution [92]. For instance, the structure of a human cytoplasmic actomyosin complex was presented at an average resolution of 3.9 Å [5]. A small 17 kDa protein (DARPin) was structurally identified and solved in near-atomic detail, ranging from 3.5 to 5 Å resolution [93]. In addition to the structural reconstruction of natural biomacromolecules, cryo-EM structure of 3D DNA origami were also reported [94,95,96]. For instance, Bai et al. reported the cryo-EM structure and a full pseudoatomic model of a discrete DNA object in 2012 [94]. Kube et al. reconstructed the structures of megadalton-scale DNA complexes with a resolution of a single nucleotide [95] (Figure 3A). These works provided a variety of stable, detailed topologies of DNA nanostructures for future application, such as assist cryo-EM structural reconstruction of proteins discussing below.

In a typical cryo-EM protocol, purified solution containing proteins is applied to a grid and then frozen in a thin layer of vitreous ice. Then, multiple cryo-EM images of individual molecule are collected, computationally sorted, and aligned. These two-dimensional (2D) images from different viewing directions can be further combined and achieve the three-dimensional (3D) reconstruction of molecular structures. However, even with the recent rapid improvements in cryo-EM techniques, there are still existing challenges in the structural reconstruction of proteins [97,98], specifically related to sample preparation [99].

First, biological macromolecules may unfold or aggregate when they hit the air–water interface, which blocks the views from different angles. Second, macromolecules may adsorb to the air–water interface in a nonrandom manner, which leads to an uneven distribution of viewing angles in the cryo-EM images that may hinder the 3D reconstruction. Finally, the high level of experimental noise caused by free movement of protein molecules can lead to errors in the a posteriori determination of the viewing angles of an individual molecule that adopts uncontrolled orientations in the ice layer, which imposes another limitation to the 3D structural reconstruction process for protein complexes. Therefore, researchers are actively seeking ways to precisely control the molecular behaviors in the cryo-EM imaging protocols involving DNA nanotechnology.

So far, several pioneering explorations have been undertaken to utilize DNA nanostructures to aid cryo-EM imaging for obtaining precise protein structures. In 2011, Selmi and coworkers proposed a self-assembled DNA nanostructure to facilitate the imaging data collection of single-particle cryo-EM [102]. The DNA array was made from a four-arm junction with a sticky end on each arm. Hybridization of complementary sticky ends could assemble these motifs into a 2D array. An ssDNA strand was modified on the array for capturing the target proteins. In this way, this DNA nanostructure can help form dense but nonoverlapping protein arrays that are suitable for imaging by cryo-EM. By using flexible lengths of linkers, the DNA templates can ensure that images of molecules with a wide range of spatial locations are obtained. The authors further applied the technique to image the structures of targets including guanine nucleotide-binding protein (Gα_i1_), G-protein-coupled membrane receptor (GPCR), and other proteins by single-particle cryo-EM.

Although the DNA array-assisted protein-capture technique provides a new strategy to facilitate structural reconstruction of single particle by confining the targets on a periodical substrate, the proteins arranged on the DNA templates were difficult to control precisely with their orientations. To address this issue, molecular supports that were constructed using DNA origami techniques have been developed. Martin et al. designed a support structure that could simultaneously control the orientations of individual particles, as well as protect the proteins from the damage of the air–water interface [9]. Figure 3B shows the design of the DNA origami molecular-support structure. A hollow pillar was formed by 82 parallel double-stranded DNA (dsDNA) helices with a height of 26 nm. The helices created a central cavity of 13.2 × 13.6 nm and an outer dimension of 26.4 × 32.7 nm. A dsDNA helix with a target protein binding sequence spanned the center of the hollow space. By tuning the locations of the central dsDNA, different relative orientations of the target protein with respect to the support structure could be induced. Specifically, the target protein rotated 34° along the axis of the central dsDNA when moving the binding sequence 1bp upward (Figure 3B, bottom). Thus, they generated five different orientations that covered the entire 180° viewing of the target protein. The device was successfully used to obtain the structure reconstruction of transcription factor p53, which is a 160 kDa tetramer with a 15 Å resolution. The presence of extra density in the front and back of the p53–dsDNA complex provided useful information about how p53 binds to dsDNA with C2 symmetry along the DNA axis, and indicated that p53 did not bind to DNA in a previously proposed arrangement with D2 symmetry. Similar to the work above, Nesrine Aissaoui et al. recently built a V-shaped DNA origami as a scaffolding molecular system to enable accurate molecular-scale positioning of proteins [103]. They utilized the template to aid the structure reconstruction of an RNAP protein with a 25 Å resolution.

Building on the work of Martin et al., Aksel and coworkers constructed DNA-origami goniometers with 14 different configurations to more precisely orient and visualize the proteins using cryo-EM [100] (Figure 3C). They used these goniometers to obtain the structure of BurrH with a further improved resolution of 6.5 Å. Specifically, this 82 kDa DNA-binding protein is known to be a target whose helical pseudosymmetry prevents accurate imaging with traditional cryo-EM. In their strategy, the goniometer was composed of a fixed chassis (gray), a programmable DNA stage containing a BurrH-binding site (magenta), and barcode domains (teal) that uniquely identified each stage-angle-related configuration of the protein (Figure 3C, upper left). The DNA stage orientation was set by the staple sequences in the anchoring regions of the chassis (dark gray) and the 5’ to 3’ direction of the scaffold sequence within the origami. The design allowed for limited swiveling of the protein rotation angle centered around a targeting angle by adding a 2 nt ssDNA strand between the DNA stage and chassis, as they claimed that excess angular rigidity may limit the particle-angle distribution and compromise reconstruction resolution. Besides, the authors designed barcode domains on the outer edges of the goniometer chassis to aid the classification each DNA stage configuration. This work demonstrated that the a priori information provided by the DNA-origami goniometers could help improve imaging resolutions of small proteins such as BurrH (6.5 Å resolution), resulting in a resolution improvement from 10.8 Å for traditional methods to 7.4 Å.

Another challenge in protein structure determination is related to the membrane proteins, since their appropriate conformations rely on lipid environment and they are unstable and insoluble upon removal from the cellular membranes. Dong and coauthors introduced a method to reconstitute a single membrane protein into a DNA barrel structure that scaffolded a lipid environment in vitro [101], as shown in Figure 3D. By careful design, α-hemolysin was incorporated into the DNA barrel in a monodispersed and nativelike state. The density map of α-hemolysin was reconstructed at a resolution of around 30 Å. The relatively low resolution may have resulted from the inevitable movement of the α-hemolysin protein relative to the DNA-origami support, since there was no physical constraint on the in-plane orientations of the α-hemolysin against the DNA-origami support. Besides, considering the limitation of the size of the cavity, the work of Dong et al. may not have been suitable to study the large complexes of membrane proteins. A lipid nanodisc with larger size may be helpful to overcome this issue. Motivated by the challenge of constructing stable lipid nanodiscs larger than 50 nm in diameter, Zhao et al. applied external DNA-origami barrels as scaffolding corrals to overcome this issue [104]. They successfully made two different-sized DNA barrels: 90 and 60 nm outer diameter to reconstitute ∼70 or ∼45 nm DNA-corralled nanodiscs (DCNDs). The DCNDs provided an excellent tool for reconstructing larger sizes of membrane protein complexes in a nativelike environment with designed composition, stoichiometry, and orientation.

## 4. Study of Protein–Protein Interactions

Protein–protein interactions (PPIs) have significant roles in physiological and pathological processes, including signal transduction, cell proliferation, growth, differentiation, and apoptosis. Aberrant PPIs are always associated with diseases such as cancer and infections. Therefore, PPIs may indicate new potential therapeutic targets for drug discovery. As a result, studying and deciphering PPIs has received increasing attention in recent years.

Owing to the addressability over nanoscale precision, DNA nanostructures and nanodevices enable precise arrangement of separate proteins and provide powerful tools and platforms for studying PPIs both in vitro and in vivo. In 2007, Williams et al. presented an approach for constructing peptide and protein nanoarrays with addressable surface features [105]. By constructing high-density peptide arrays on the nanoscale, this work provided a potential platform for studying PPIs, since the nanoarrays could firmly anchor individual protein molecules in a well-controlled spatial dimension. However, the authors did not apply the platform to study practical models of PPIs in this work.

In 2020, Rosier et al. presented a DNA origami-based synthetic apoptosome to analyze signaling proteins in vitro [106] (Figure 4A). Apoptosome is a multiprotein complex that regulates apoptosis by colocalizing multiple caspase-9 monomers. By using rectangle DNA origami as an artificial apoptosome to assemble individual caspase-9 monomers with absolute control over their position, their studies revealed that caspase-9 activity was induced by proximity-driven dimerization. They were also able to study the effect of higher-order clustering on caspase-9 activity by constructing three- and four-enzyme systems. The results suggested that a multivalent catalytic effect led to enhanced activity in caspase-9 oligomers.

Knowing the dynamics of antibody binding to variably distributed antigens would be highly desirable for deeper understanding of the initiation of immune responses, and for rational vaccine design as well. Shaw et al. proposed a patterned surface plasmon resonance (PSPR) method to characterize the dynamic interplay between the antibody structural flexibility and binding to cognate antigen patterns in real time [108]. By displaying precise nanoscale patterns of antigens on a DNA-origami nanostructure, they found that the binding affinities of antibodies changed with the antigen distances, with a distinct preference for antigens separated by approximately 16 nm. In addition to the DNA-nanotechnology-based in vitro study of antigen–antibody binding, Veneziano et al. studied the independent roles of antigen spatial arrangement, antigen copy number, and additional parameters (including antigen affinity and structural rigidity) on B-cell activation using DNA-origami-based 40 nm viral-like nanoparticles [109]. While displaying immunogens at high density have all been shown to strongly initiate early B-cell signaling, their studies revealed that B-cell activation was maximized by five antigens maximally spaced on the viral-like nanoparticles’ surfaces. These results provided useful design principles that may be used to stimulate B-cell responses in humoral immunity.

Kostrz et al. introduced a junctured-DNA tweezer that was composed of three chemically linked linear segments of double-stranded DNA (dsDNA), in which two DNA ends (termed tips) could be pulled apart with a controlled force [107] (Figure 4B). These tips could be specifically engrafted with molecular partners of interest. By monitoring the extension of the DNA scaffold using magnetic tweezers, the researchers could specify whether the two molecules of interest located at the tips were associated or dissociated. This tool provided an accurate and model-independent measurement of the lifetime of protein–rapamycin interactions. The determined rate constants were in good agreement with those reported in other literature. Additionally, this method only needs less than 1% of the amount of protein required for traditional methods.

Although there are many well-constructed methods for studying PPIs in vitro, it is still difficult to obtain PPIs in situ, such as on cell membranes or inside the cells. Ambrosetti et al. developed a non-microscopy-based method for ensemble analysis of membrane proteins’ interactions in situ [10] (Figure 4C). Their method was based on the use of a DNA nanostructure (termed NanoComb) to locally encode membrane protein spatial distribution information into a newly synthesized sequence that could be read later by DNA sequencing. The authors successfully applied this to characterize the nanoenvironments of Her2 on SKBR3 and MCF7 cell surfaces. This strategy showed the potential to enable the high-throughput analysis of PPIs in situ without microscopy.

## 5. Measurement of Molecular Forces

Forces in biological systems are typically characterized at the single-molecule level with optical tweezers, magnetic tweezers, atomic force microscopy [4], or optical trapping (OT). However, these techniques are limited by low throughput, and sometimes require complicated experimental preparation. In 2016, Iwaki et al. developed a programmable DNA origami nanospring to overcome these issues [110]. The nanospring exhibited a relatively small spring constant of 0.012 ± 0.002 pN nm^−1^ and a length of 841 nm at 2 pN of force, which was an ideal nanospring for studying myosin VI. They observed force-induced transitions of myosin VI heads from nonadjacent to adjacent binding, which corresponded well to previously predicted adapted roles for low-loading and high-loading transport.

In addition to the study of molecular forces of motor proteins, Funke et al. developed a DNA origami-based force spectrometer that could measure the forces between nucleosomes [111] (Figure 5A). They used a DNA-based positioning device to place two nucleosomes close to each other in a defined orientation. The nanodevice constrained the relative motion of the nucleosomes, and featured an effective spring that counteracted the attractive interactions between the nucleosomes. The spring converted the positioning device into a force spectrometer. By sampling the frequency at particular conformations by single-particle transmission electron microscopy (TEM), the nucleosome–nucleosome distances were measured by the spectrometer. They also used a set of dyes to report the conformation via a FRET signal. This strategy provided a powerful tool for studying molecular forces via imaging techniques.

Similar to the aforementioned method, Nickels et al. introduced a self-assembled nanoscopic force clamp built from DNA that allowed massive parallelization of analysis [112]. As shown in Figure 5B, the ssDNA spring that was part of the long scaffold strand spanned the gap of a bracket-shaped DNA origami clamp. The ssDNA sections acted as entropic springs, and could exert controlled tension in the low piconewton range. Conformational transitions of the ssDNA springs could be monitored with a single-molecule FRET system (Figure 5B, right). This work successfully generated force spectroscopy data in a dynamic range of 0 to 12 pN, presenting a method for studying molecular forces with high data throughput and simple operation.

Mechanical stimuli are critical for many cellular processes, including migration, proliferation, and differentiation [113,114]. Salaita’s group developed DNA-based tension probes to reveal the piconewton forces exerted by cell surface receptors [115,116,117,118,119]. For instance, they used molecular tension probes to reveal integrin forces during early cell adhesion [115]. As shown in Figure 5C, the probes consisted of three ssDNA strands assembled through hybridization of a 21 bp handle: a stem-loop DNA hairpin (black), a peptide-modified ligand strand conjugated to a fluorophore (green), and a surface-anchor strand conjugated to a quencher (blue). Thus, a sufficient force led to the hairpin unfolding, accompanied by a drastic increase in fluorescence intensity. In addition, the sequences of the DNA stem could be used to rationally tune its force-response function. In 2019, they further improved the DNA-based tension probes that could store mechanical information so that imaging short-lived forces transmitted by low-abundance receptors were realized [119] (Figure 5D). While efforts had been made to develop a DNA-origami-based molecular force analysis method, it was noticed that the unraveling of DNA-origami structures under force may occur during the study of molecular force. Engel et al. explored the force response of DNA origami in comprehensive detail [120]. Their work studied the stability of DNA nanostructures under external forces, and what design principles can be applied to enhance the structural stability, which may be helpful for developing related applications.

## 6. Conclusions and Perspectives

Recent advances have demonstrated the feasibility and advantage of obtaining precise molecular information via DNA nanotechnology, which shows incomparable potentials in applications that require control over nanoscale features. As we mentioned above, DNA-based nanoarchitectures and probes have been applied to facilitate the structural reconstruction of protein, in situ study of PPIs, and the revealing of molecular forces. With the decrease of the cost of DNA synthesis and modification, as well as the progress in design, characterization, and functionalization of DNA nanostructures, there will definitely be more and more DNA-based devices applied to explore unknown facts in science. While DNA nanotechnology has presented tremendous opportunities in single-molecule research, this field is still facing many challenges. For instance, the DNA-origami-based goniometers or scaffolds used for structural reconstruction of proteins can only be applied to proteins that can bind to a specific DNA domain, which greatly limits the versatility of the strategy. To prepare goniometers with different barcodes, multiple types of origami need to be designed, folded, and purified, which is laborious and slow. Hopefully, the design [52,53,54] and synthesis of DNA nanostructures can be automatic in the future, providing a batch of ready-to-use tools for researchers. However, the instability of DNA nanostructures in physiological conditions limits their further application in vivo [121]. In addition, relatively high concentrations of cations (i.e., Mg^2+^) needed for maintaining the intactness of DNA nanostructures may more or less affect the interaction of biomolecules. Thus, efforts are still required to improve the stability of DNA nanostructures towards more biological applications [122,123,124,125]. However, considering that DNA plays an irreplaceable role in nanoscale manipulation, this field will definitely continue surging in the coming future.

## Figures and Tables

**Figure 1 membranes-11-00683-f001:**
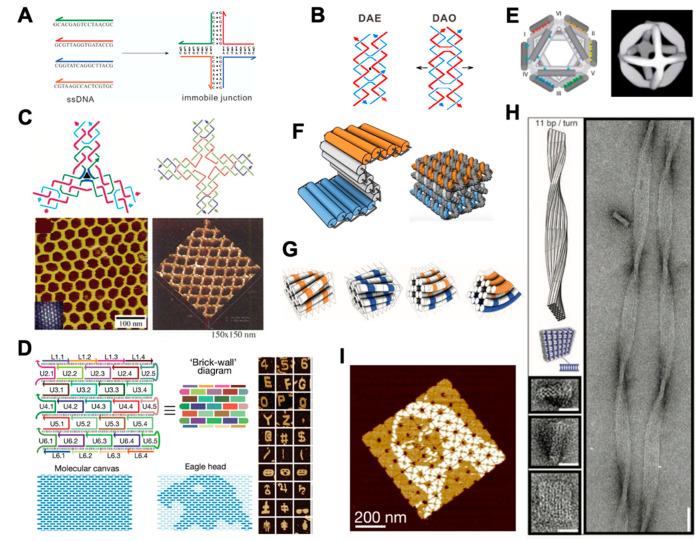
DNA nanostructures created by DNA tile and DNA origami assembly. (**A**) Artificial immobile junction assembled from four ssDNA strands, adapted with permission from [37]. (**B**) Double crossover structures, adapted with permission from [38]. (**C**) 2D DNA crystals made from DNA tiles with three and four arms, adapted with permission from [39,40] (**D**) DNA bricks for the assembly of 2D objects, adapted with permission from [11]. (**E**) An earlier 3D octahedron created prior to the birth of DNA origami, adapted with permission from [41]. (**F**) Cylinder model and atomistic DNA model of a honeycomb-pleated origami, adapted with permission from [27]. (**G**) Cylinder model of the twisted and curved 3D DNA origami, adapted with permission from [17]. (**H**) Models and TEM images of a twisted 10 × 6-helix DNA bundle, adapted with permission from [17]. (**I**) An 8 × 8 array of DNA origami with an example pattern of Mona Lisa, adapted with permission from [42].

**Figure 2 membranes-11-00683-f002:**
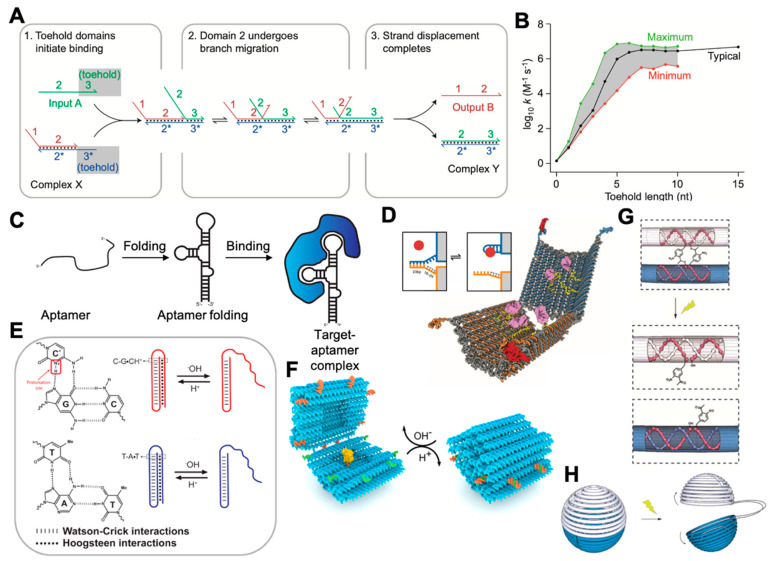
Dynamic DNA nanotechnology and its application in constructing dynamic devices. (**A**) An example of TMSD, adapted with permission from [90]. (**B**) The kinetics of TMSD can be tuned by changing the length and sequence of the toehold domain, adapted with permission from [90]. (**C**) Schematic representation of aptamer binding to a target protein, reproduced from [91] under terms of the CC BY 4.0 license. (**D**) An DNA nanorobot that can be opened by receptor-induced binding with aptamers, adapted with permission from [71]. (**E**) pH-triggered nanoswitches that form an intramolecular triplex structure through the formation of a Watson–Crick (dashed) pH-insensitive hairpin and a second Hoogsteen (dots) pH-sensitive hairpin, adapted with permission from [80]. (**F**) Hoogsteen base-pairing-based box that closes in low-pH conditions, adapted from [75] under the terms of the Creative Commons CC BY license. (**G**,**H**) Ortho-nitrobenzyl functionalized DNA capsule that opens upon exposure to light, adapted with permission from [81].

**Figure 3 membranes-11-00683-f003:**
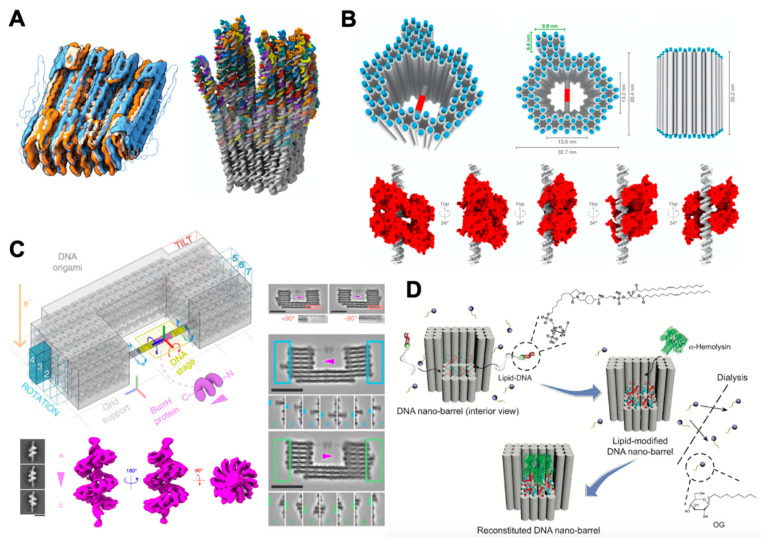
DNA nanotechnology applied in the structural reconstruction of proteins. (**A**) Overlay of the cryo-EM maps of the 48-helix-brick (left) and atomic models of a twist tower derived from fitting to the cryo-EM map (right), adapted from [95] under the terms of the Creative Commons CC BY license. (**B**) Perspective view of the DNA origami molecular support (top) and illustration of five different settings for the p53 binding on the central dsDNA helix (bottom), adapted with permission from [9]. (**C**) Illustration of the DNA origami chassis (top left) and its TEM (right); 3D reconstruction of BurrH is shown on bottom left, adapted with permission from [100]. (**D**) Views of the design of the DNA nanobarrel with a central pore and reconstruction of a-hemolysin into the DNA nanobarrel through lipid–protein interaction, adapted with permission from [101].

**Figure 4 membranes-11-00683-f004:**
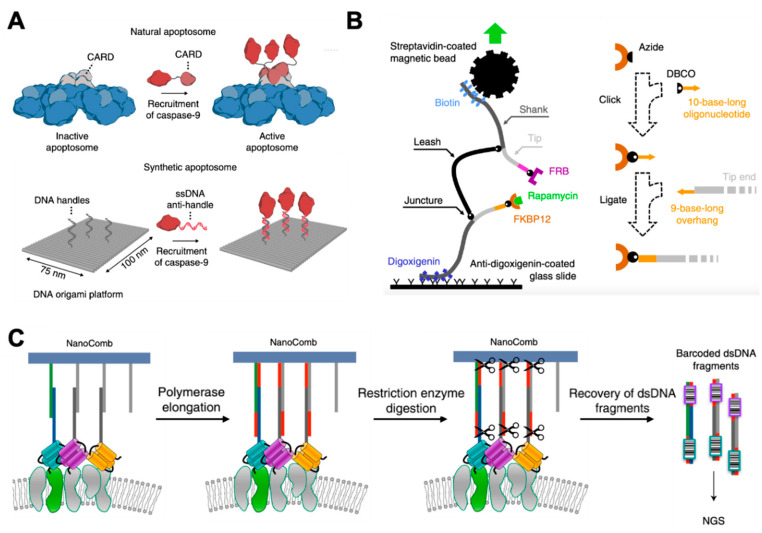
Study of PPIs based on DNA nanotechnology methods. (**A**) Self-assembled 2D peptide nanostructure, adapted with permission from [106]. (**B**) Structure of the junctured-DNA tweezer (left) and the strategy used to attach a given protein at a given tip (right), adapted with permission from [107]. (**C**) Schematic of the DNA NanoComb method, adapted with permission from [10].

**Figure 5 membranes-11-00683-f005:**
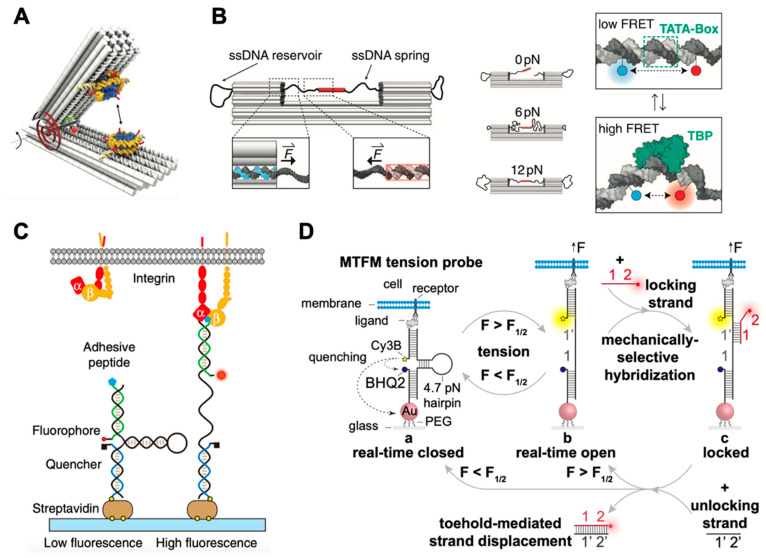
Measurement of molecular forces via DNA nanotechnology. (**A**) Schematic of the DNA force spectrometer featuring a spring-loaded hinge with two attached nucleosomes, reproduced from [111] under terms of the CC BY 4.0 license. (**B**) Structure of the DNA-origami force clamp, with ssDNA reservoirs located on each side of the clamp (left); individual origami samples were assembled for each constant-force variant (middle); TBP-induced DNA bending under force were monitored by FRET (right), adapted with permission from [112]. (**C**) Schematic of the integrin tension sensor, adapted with permission from [115]. (**D**) Schematic depicting the concept of mechanical information storage, adapted from [119] under terms of the CC BY license.

## Data Availability

Not applicable.

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
