# Peer review of "Obtaining Precise Molecular Information via DNA Nanotechnology"

_membranes, 2021, doi:10.3390/membranes11090683_

Round 1

Reviewer 1 Report

The Review by Tang and Han, is an interesting report of the state of the art DNA nanotechnology-based methodologies to study biomolecular interactions. While this review will certainly be of interests for readers that are outside of the DNA nanotechnology field, I feel that there is few important citations and aspects of DNA nanotechnology that are missing and that should be added before publications. Here are my major and minor concerns:

Major points:

  • The introduction section would benefit from more general references to the DNA nanotechnology field. There is only 7 citations in this entire section and only 3 are related to the DNA nanotechnology field. It would be nice to have references that describe the use of DNA as a building block/materials in this section and not only references about DNA origami.
  • Overall, I think this review would benefits from adding more citations that pertains to the applications of DNA nanotechnology in the different sections. Below I will highlight some important citations that I think are missing.
  • In the paragraph 2 which describe the field of DNA nanotechnology, the unique properties of DNA as a building blocks are not clearly explained. The authors only mentioned that "the Thermodynamics of DNA hybridization have been well characterized". It would be nice to explain here in details the properties of DNA (addressability, sequence specificity, Watson-Crick base pairing, programmability, biocompatibility, stability, mechanical properties and structural properties). 
  • In the same paragraph the authors mention that DNA is the "one of the most easily obtained building blocks currently available". Is that really true? DNA is still expensive and maybe it would be better to comments on its properties versus other polymeric materials instead.
  • In the captions it would be preferable to have the references listed in each part instead of having them all at the end of the captions. It would be easier for the reader to find the right references.
  • On the paragraph 2.1, the authors are not citing anything about recent automated design strategy from lab such as work from the Hogberg and the Bathe group. I think a paragraph that describes these methods is required as these designing software are helping the field to get more accurate and structurally rigid structure for further analysis.
  • In the paragraph 2.2 the authors state that the "reactions can be quantitatively controlled over a factor of 10^6", could explain that a little more to help the reader understand the importance of that fact.
  • In the section 3, the authors are not discussing at all the progress made in imaging the DNA origami first. Even if it is not the point of this review this is crucial for studying protein on DNA origami. If the structure are not rigid enough or not adapted for Cryo-EM it will be difficult to study protein on it. Cite for example work from the Dietz group (Bai et al., PNAS 2012)
  • At the end of the section 3, the authors should discuss more the concept of lipid nanodisk reconstituted with DNA origami to study membrane protein and for example mention the work from Zhao et al (2018, JACS, 10.1021/jacs.8b04638).
  • In the section 4, while biophysics characterization was mentioned earlier, the authors are not discussing any of the recent papers that use SPR to study biomolecular interactions with DNA origami and protein complex. For example Shaw et al 2019 in Nat. Nanotech should be discussed here.
  • In the conclusion and perspectives, the authors state that "Hopefully, the design and synthesis of DNA nanostructures could be automatically in the future,...". Here the authors are not acknowledging all the recent effort made by the field to propose solution and software for the automated design of DNA origami. It pertains to my previous point where I asked the authors to discuss these software in the context of their review. Please modify the conclusion.

Minor points

  • In introduction: The authors should say: DNA (deoxyribonucleic acid) molecule is the...
  • Section 2 line 3, it is base pairing and not base paring.
  • In captions of the figure the authors use capital A, B, C... but use small letter in the figure.
  • Page 5 line 6, the word "intact" should be replaced by interact. Same line the sentence is not clear and should be changed
  • Page 6, second the last line, the sentence "and have the advantages of rapid and remotely controllable is not clear" do you mean the advantages to be rapidly and remotely controllable?
  • Page 7: Please explain a little more the concept of azobenzene actuation as some readers might not now what it means and how it works.
  • Page 8, "different angels" should be replaced by different angles
  • Posteriori (page 7)and priori (page 10) should be replaced by a posteriori and a priori
  • Page 7: the last sentence that finish by starts to be involved in should be changed. DNA nanotechnology is not involve but can be useful or enable new study...

Author Response

Please see the attached report

Reviewer 2 Report

Overall this review paper gives a broad introduction to DNA nanotechnology and a more detailed specific review of applications to the study of biomolecular systems, such as protein structure determination and protein-protein interactions. This focus area is particularly interesting, and recent progress justifies a focused review. As far as I am aware, the scope is distinct to other recent reviews in this area, see specific notes below. The paper is well written and contains an appropriate amount of detail. In the introduction and conclusion there are some general statements, it would strengthen the paper to add example references for these points, as noted below. In the body of the text, referencing was generally good, some examples of key references that should be added are given below. 

I recommend changes on the following minor points:
Page 2 "researchers are also able to reconstruct the structures of biomolecules such as proteins at an atomic level resolution and accurately investigate intermolecular interactions such as protein-protein interactions" 
Is useful to cite relevant examples here 

Page 2 "transforming our understanding of complicated biological processes"
Add a reference example for this

Page 2 - final paragraph 
Cite some more general reviews of the area, such as:
Seeman, N.C., and Sleiman, H.F. (2018). DNA nanotechnology. Nat Rev Mater 3, 17068.
Pinheiro, A.V., Han, D., Shih, W.M., and Yan, H. (2011). Challenges and opportunities for structural DNA nanotechnology. Nature Nanotechnology 6, 763–772.
Cite some recent reviews of related areas but with different focus:
Hernandez-Garcia, A. (2021). Strategies to Build Hybrid Protein–DNA Nanostructures. Nanomaterials 11, 1332.
Stephanopoulos, N., and Šulc, P. (2021). DNA Nanodevices as Mechanical Probes of Protein Structure and Function. Applied Sciences 11, 2802.

Figure captions (all)
- citations should be changed to indicate which subfigure belongs to which paper, rather than all be listed at the end of the caption 

Page 5, Add cadnano reference 
Douglas, S.M., Marblestone, A.H., Teerapittayanon, S., Vazquez, A., Church, G.M., and Shih, W.M. (2009). Rapid prototyping of 3D DNA-origami shapes with caDNAno. Nucleic Acids Res 37, 5001–5006.

Page 6 "relatively slow in kinetics", Add some specific examples of rate constants for TMSD and other switching types

Page 8 "complexes with near-atomic resolution" give reference and quote the specific resolution achieved
Generally, the background section on cryo-EM techniques is helpful, but needs more references. For example, a reference to a review paper of cryo-EM techniques would be useful here

Page 11 "Dong and coauthors introduced a method to reconstitute single membrane protein..." This particular example will be of particular interest to the audience of the membranes journal. It is not clear from the text here whether the resolution achieved  was useful or if this method requires improvement. Please clarify this

Page 12: Protein-protein interactions section is missing the following two recent examples, should be included here:
Rosier, B.J.H.M., Markvoort, A.J., Gumí Audenis, B., Roodhuizen, J.A.L., den Hamer, A., Brunsveld, L., and de Greef, T.F.A. (2020). Proximity-induced caspase-9 activation on a DNA origami-based synthetic apoptosome. Nature Catalysis 3, 295–306.
Veneziano, R., Moyer, T.J., Stone, M.B., Wamhoff, E.-C., Read, B.J., Mukherjee, S., Shepherd, T.R., Das, J., Schief, W.R., Irvine, D.J., et al. (2020). Role of nanoscale antigen organization on B-cell activation probed using DNA origami. Nat. Nanotechnol. 15, 716–723.

Page 14 'Forces in biological systems...'. 
This section should include the following reference on force-sensing with DNA origami:
Iwaki, M., Wickham, S.F., Ikezaki, K., Yanagida, T., and Shih, W.M. (2016). A programmable DNA origami nanospring that reveals force-induced adjacent binding of myosin VI heads. Nature Communications 7, 13715.
It would also benefit from a discussion of limitations of DNA origami unravelling under force, for example:
Engel, M.C., Smith, D.M., Jobst, M.A., Sajfutdinow, M., Liedl, T., Romano, F., Rovigatti, L., Louis, A.A., and Doye, J.P.K. (2018). Force-Induced Unravelling of DNA Origami. ACS Nano 12, 6734–6747.

Page 16, "instability of DNA nanostructure in physiological conditions limits..."
Include reference for this: 
Hahn, J., Wickham, S.F.J., Shih, W.M., and Perrault, S.D. (2014). Addressing the Instability of DNA Nanostructures in Tissue Culture. ACS Nano 8, 8765–8775.

Page 16 "efforts are still required to improve the stability of DNA nanostructures towards more biological applications." 
Add examples of progress in this direction, such as:
Ponnuswamy, N., Bastings, M.M.C., Nathwani, B., Ryu, J.H., Chou, L.Y.T., Vinther, M., Li, W.A., Anastassacos, F.M., Mooney, D.J., and Shih, W.M. (2017). Oligolysine-based coating protects DNA nanostructures from low-salt denaturation and nuclease degradation. Nature Communications 8, 15654.
Gerling, T., Kube, M., Kick, B., and Dietz, H. (2018). Sequence-programmable covalent bonding of designed DNA assemblies. Science Advances 4, eaau1157.
Anastassacos, F.M., Zhao, Z., Zeng, Y., and Shih, W.M. (2020). Glutaraldehyde Cross-Linking of Oligolysines Coating DNA Origami Greatly Reduces Susceptibility to Nuclease Degradation. J. Am. Chem. Soc. 142, 3311–3315.

Typographical errors to correct:

Page 2 "Benefits from these single-molecule techniques, ". Grammar is a bit confusing in this sentence
Page 2 "these assembly accuracy", 'these' should be 'the'

Page 5, paragraph beginning "DNA origami was first developed by Rothemund..." is a bit too long, split into two paragraphs. 

Page 8 "from different angels" typo, should be angles

Page 8 "imaging protocols where DNA nanotechnology starts to be involved in." Grammar not clear, could correct to 'imaging protocols involving DNA nanotechnology'

Reviewer 3 Report

This well-written review paper by Tang and Han concisely explains basic background of structural DNA nanotechnology and its application to biosensing/analysis of protein structures, protein-protein interactions and inter-molecular force. Most of the epoch-making works in the field are mentioned and briefly introduced in the text. Though it is already sufficiently attractive for the readers in the current form, adding two following references to the manuscript would make the paper more impartial and informative:

  1. The following study might be the first successful dynamic DNA origami implemented with various bio-related interactions such as target-aptamer binding and non-canonical DNA structure formation triggered by pH or ionic-strength changes. It should be referred around ref. 39: Kuzuya, A.; Sakai, Y.; Yamazaki, T.; Xu, Y.; Komiyama, M. Nanomechanical DNA origami ‘single-molecule beacons’ directly imaged by atomic force microscopy. Nat. Commun. 20112, 449.
  1. For section 3, some DNA origami structures are now listed in PDB. Following ground-breaking study should be referred in the text: Kube, M. et al. Revealing the structures of megadalton-scale DNA complexes with nucleotide resolution. Nat. Commun. 202011, 6229.

Author Response

Please see the attach file

Round 2

Reviewer 1 Report

The authors addressed the major comments of the reviewer. I only have few minor comments before publication.:

1- The new paragraph 2 named DNA nanotechnology: The title is a little misleading since it is mainly about the structure of DNA, so it would be better if the authors change the title to: "Structural features of DNA" or something similar.

2- In the same paragraph, the authors are not citing any publications, they should at least have the citation #30 listed in this paragraph and probably some more general references about DNA and few were they mention that: "In the past few decades, the thermodynamics of DNA hybridization have been well characterized". This is a review so all the statements need to be associated with references.

3- In page 6, in the new paragraph about automated design the authors have chosen to present vHelix and PERDIX specifically. Why did they choose these two software only. There is many other software now available for automated design of Dx-Tile based structures, 6HB structures,... It might be good to cite more references here.

4- In the same paragraph in page 6, Cadnano should be mentioned along with Tiamat and scadnano.
